# *Mycobacterium* dormancy and antibiotic tolerance within the retinal pigment epithelium of ocular tuberculosis

Rachel Liu,[1] Joshua N. Dang,[2] Rhoeun Lee,[1,3] Jae Jin Lee,[1] Niranjana Kesavamoorthy,[2] Hossein Ameri,[2] Narsing Rao,[2] Hyungjin Eoh[1,2]

**ABSTRACT** Tuberculosis (TB) is a leading cause of death among infectious diseases worldwide due to latent TB infection, which is the critical step for the successful pathogenic cycle. In this stage, *Mycobacterium tuberculosis* resides inside the host in a dormant and antibiotic-tolerant state. Latent TB infection can also lead to multisystemic diseases because *M. tuberculosis* invades virtually all organs, including ocular tissues. Ocular tuberculosis (OTB) occurs when the dormant bacilli within the ocular tissues reactivate, originally seeded by hematogenous spread from pulmonary TB. Histological evidence suggests that retinal pigment epithelium (RPE) cells play a central role in immune privilege and in protection from antibiotic effects, making them an anatomical niche for invading *M. tuberculosis*. RPE cells exhibit high tolerance to environmental redox stresses, allowing phagocytosed *M. tuberculosis* bacilli to maintain viability in a dormant state. However, the microbiological and metabolic mechanisms determining the interaction between the RPE intracellular environment and phagocytosed *M. tuberculosis* are largely unknown. Here, liquid chromatography-mass spectrometry metabolomics were used to illuminate the metabolic state within RPE cells reprogrammed to harbor dormant *M. tuberculosis* bacilli and enhance antibiotic tolerance. Timely and accurate diagnosis as well as efficient chemotherapies are crucial in preventing the poor visual outcomes of OTB patients. Unfortunately, the efficacy of current methods is highly limited. Thus, the results will lead to propose a novel therapeutic option to synthetically kill the dormant *M. tuberculosis* inside the RPE cells by modulating the phenotypic state of *M. tuberculosis* and laying the foundation for a new, innovative regimen for treating OTB.

**IMPORTANCE** Understanding the metabolic environment within the retinal pigment epithelium (RPE) cells altered by infection with *Mycobacterium tuberculosis* and mycobacterial dormancy is crucial to identify new therapeutic methods to cure ocular tuberculosis. The present study showed that RPE cellular metabolism is altered to foster intracellular *M. tuberculosis* to enter into the dormant and drug-tolerant state, thereby blunting the efficacy of anti-tuberculosis chemotherapy. RPE cells serve as an anatomical niche as the cells protect invading bacilli from antibiotic treatment. LC-MS metabolomics of RPE cells after co-treatment with $H_2O_2$ and *M. tuberculosis* infection showed that the intracellular environment within RPE cells is enriched with a greater level of oxidative stress. The antibiotic tolerance of intracellular *M. tuberculosis* within RPE cells can be restored by a metabolic manipulation strategy such as co-treatment of antibiotic with the most downstream glycolysis metabolite, phosphoenolpyruvate.

**KEYWORDS** *Mycobacterium tuberculosis*, ocular tuberculosis, metabolomics, chemotherapy

Tuberculosis (TB) is a global infectious disease with high mortality and morbidity caused by infection with *Mycobacterium tuberculosis*, primarily affecting the lungs

Address correspondence to Hyungjin Eoh, heoh@usc.edu.

The authors declare no conflict of interest.

See the funding table on p. 12.

—a condition known as pulmonary TB. *M. tuberculosis* can also affect other organs, leading to extrapulmonary TB, which accounts for approximately 16% of total TB cases reported annually (1, 2). Extrapulmonary TB can arise from either primary infection or secondary infection, where the pathogen spreads from the primarily affected organs (3). Secondary infections typically occur due to the hematogenous or lymphatic spread of *M. tuberculosis* bacilli from the lungs, which evades the host immune system or antibiotic effects by entering a dormant state (4, 5). The bacilli can become pathogenic through opportunistic reactivation (6). A significant challenge for an efficient control for extrapulmonary TB cases is the lack of proper diagnostic methods and treatment regimens. The World Health Organization recommends a 6-month 2RHZE-4RH regimen comprising rifampicin (R), isoniazid (H), pyrazinamide (Z), and ethambutol (E) for the initial 2 months, followed by RH for the subsequent 4 months (7, 8). The necessity for such an extended treatment course is primarily attributed to the well-documented mycobacterial dormancy and the high antibiotic tolerance associated with latent TB infection. Conventional TB chemotherapy struggles to eradicate the bacilli effectively, and thus, surviving bacilli can regrow when the antibiotic effects wane or when resistant mutants emerge. Consequently, there is a pressing need for the development of appropriate treatment strategies to combat the dormant *M. tuberculosis* bacilli present in the cases of extrapulmonary TB.

Ocular tuberculosis (OTB) represents a form of extrapulmonary TB that should never be underestimated if considering its potential to cause significant visual loss in affected patients (9–11). Similar to other extrapulmonary TB cases, OTB can arise from primary infection within the ocular tissues or more commonly as a secondary infection disseminated from the lungs via the bloodstream. OTB leads to inflammation of the uvea and retina, resulting in TB uveitis, which stands as the most prevalent manifestation of OTB and is a key contributor to ocular inflammatory diseases (9, 12, 13). Diagnosing OTB poses a formidable challenge, as the majority of OTB patients do not exhibit pulmonary TB symptoms, and established gold standards for investigations and diagnostic criteria are currently lacking. Furthermore, OTB is typically paucibacillary and hence is difficult to diagnose the cases through conventional methods like smear microscopy, culture, or PCR. Consequently, timely diagnosis of OTB is often significantly delayed, with diagnosis remaining presumptive. Therefore, there is an urgent need to enhance our understanding of OTB pathogenesis and the interaction between ocular tissues and invading *M. tuberculosis* to enable the development of innovative treatment strategies.

The retinal pigment epithelium (RPE) cells form a layer that lines the outer retina, separating the photoreceptors and neuroretina from the choroid and the systemic vasculature, thereby establishing the blood-retina barrier (14, 15). The vasculature at the choroid system carries high oxygen tension passing through the RPE layer toward the retina. As RPE cells harbor a large number of mitochondria, the mitochondrial complexes generate great levels of oxidative stress by producing superoxide, $H_2O_2$, and hydroxyl radical (16). The intrinsic environment within RPE cells is thus high in oxidative stress, and RPE cells have an important role in meeting the metabolic demand (17). The RPE cells play a crucial role in maintaining this barrier by serving as a physical lining through tight junctions. They facilitate the diffusion and transport of nutrients between the systemic circulation and the retina, while also regulating local immune system activity through the release of inflammatory cytokines (18–20). Essentially, RPE cells are responsible for the immune privilege of ocular tissues, making them a primary niche where invading pathogens reside. Indeed, previous studies investigating the cases of OTB uveitis have indicated that *M. tuberculosis* mostly localizes within the RPE cells followed by pathogenic reactivation often when the host's immune system is compromised (11, 21–23). This finding underscores the significance of RPE cells in the pathogenesis of OTB and highlights their involvement in the disease progress.

The present study investigated the metabolic and biochemical states of the RPE intracellular environment that induce invading *M. tuberculosis* bacilli to enter a dormant and antibiotic-tolerant state. To explore the causality, liquid chromatography-mass

spectrometry (LC-MS) metabolomics and qRT-PCR were employed to track the metabolic changes specific to RPE cells in response to environmental redox stresses and *M. tuberculosis* infection. Additionally, we also revealed that mycobacterial dormancy is linked to a heightened level of antibiotic tolerance against key first-line TB antibiotics. The findings of this study laid the groundwork for developing a novel treatment approach by modulating the intracellular metabolic state of *M. tuberculosis* through co-administration of phosphoenolpyruvate (PEP), the most downstream glycolysis intermediate of *M. tuberculosis*, along with known TB antibiotics. Treatment with PEP was shown to prevent the induction of *M. tuberculosis* dormancy within RPE cells, synthetically improving antibiotic susceptibility. This study not only elucidates the mechanistic underpinnings of RPE cellular metabolic remodeling in mediating mycobacterial dormancy but also introduces a fresh strategy for enhancing the efficacy of OTB treatment.

## RESULTS AND DISCUSSION

### Establishment of an *in vitro* RPE culture model to form dormant *M. tuberculosis* infection

A zebrafish model infected with red-fluorescent *Mycobacterium marinum* enabled intravital visualization of host-pathogen interactions (24). The bacilli were observed within the RPE cells. RPE cells represent a physiological niche where *M. tuberculosis* bacilli reside in a dormant state (22, 25). Despite the intensive studies, there is still a significant gap of knowledge regarding how intracellular *M. tuberculosis* bacilli maintain viability in a dormant state, and this mycobacterial dormancy in OTB cases is associated with antibiotic tolerance. To investigate this, we established an *ex vivo* RPE culture system under oxidative stress induced by treatment with $H_2O_2$. This system mimicked the physiological intracellular environment associated with mycobacterial dormancy. $H_2O_2$ is a well-known natural source of reactive oxygen species (ROS) within bacterial pathogens (26–28). Given that the RPE cellular layer serves as a frontline defense mechanism against the high redox stresses from hematogenous ROS, we hypothesized that intracellular *M. tuberculosis* bacilli within RPE cells encounter high levels of oxidative stress, leading to the induction of mycobacterial dormancy. To test the hypothesis, we utilized $H_2O_2$ as a source of oxidative stress and investigated a range of $H_2O_2$ concentrations optimal for mimicking physiological intracellular stress levels in RPE cultures post-infection with H37Ra bacilli at a multiplicity of infection (MOI) of 10:1 (29). We monitored bacterial growth kinetics by counting colony-forming units (CFU). Under a resting condition without $H_2O_2$ treatment, intracellular H37Ra inside RPE cells replicates at a rate similar to that in THP-1 macrophages for initial 5 days, albeit a significantly longer lag phase (Fig. 1A). Treatment with increasing doses of $H_2O_2$ progressively slowed down the replication rates of H37Ra within RPE cells up to 100 µM, with minimal impact on bacterial growth within THP-1 macrophages or under an *in vitro* condition (Fig. 1A; S1A). The growth kinetics revealed that the mycobacterial dormancy state in the RPE cells needs the environmental redox stresses, and the redox stress levels within RPE cells in response to the same $H_2O_2$ concentrations were significantly greater than those in THP-1 macrophages.

To further examine if the slowed H37Ra replication was attributed to the greater levels of redox stresses arising from environmental ROS, we measured the amount of ROS biosynthesized by either RPE cells or THP-1 macrophages after treatment with the same concentrations, 50 or 100 µM, of $H_2O_2$. Notably, ROS levels in the RPE cells significantly increased after day 2 post-treatment with $H_2O_2$, while no increase of ROS in the THP-1 macrophages was observed (Fig. S1B). The induced ROS level was not associated with loss of RPE cell viability (Fig. S1A, right panel). Furthermore, the slowed replication rate of intracellular H37Ra within RPE cells was nearly fully restored by the treatment with 10 mM thiourea, a chemical known as an antioxidant, becoming the growth rate at a level similar to that observed in THP-1 macrophages (Fig. S1C). Thus, *in vitro* RPE cell culture treated with 50 µM $H_2O_2$ was the condition that successfully

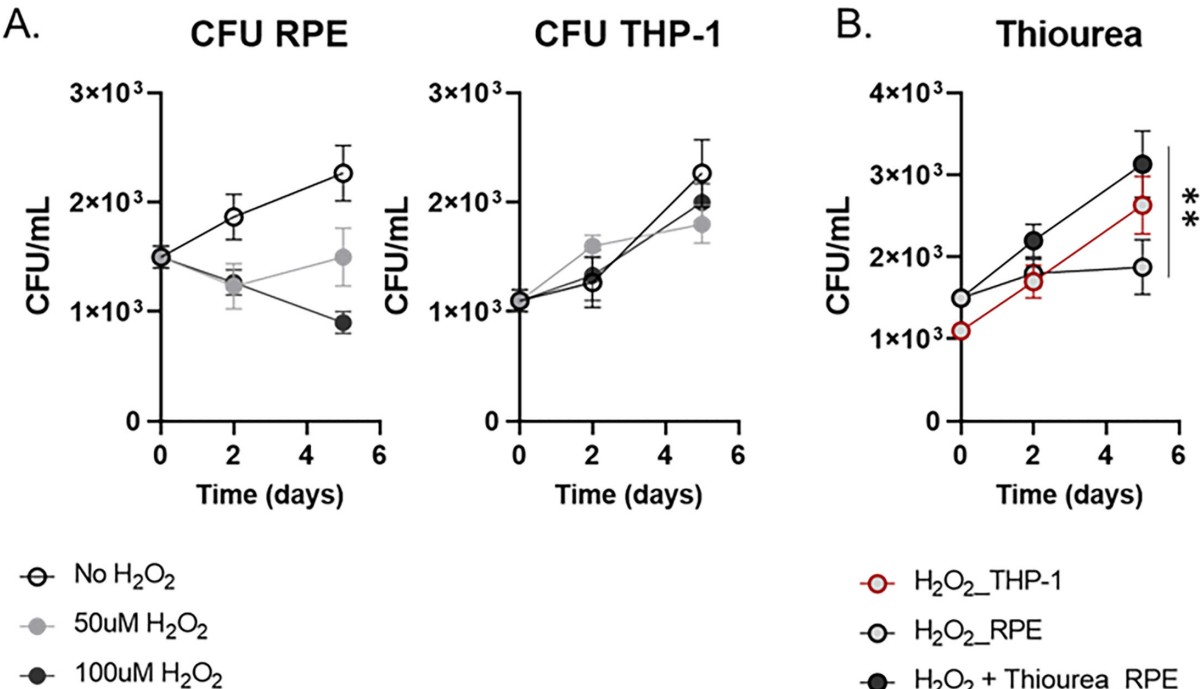

**FIG 1** Mycobacterial dormancy in RPE cells after exposing to the environmental redox stress. (A) CFU viability of intracellular H37Ra following infection of RPE cells or THP-1 macrophages with the treatment of 50 or 100 µM $H_2O_2$. (B) The effect of additional supplementation with 10 mM thiourea on CFU viability of H37Ra following treatment with 50 µM $H_2O_2$. Data points show the average of experimental triplicates ± SEM. **, $P < 0.05$ by Student's $t$ test.

generated the mycobacterial dormancy similar to the physiological condition frequently observed in OTB cases.

## Mycobacterial dormancy within the *in vitro* RPE culture system

Since dormant *M. tuberculosis* bacilli are known to accumulate triacylglycerol (TAG) at the cell wall as a critical indicator of mycobacterial dormancy (30–32) (Fig. 2A), we used our *in vitro* RPE culture system to collect H37Ra bacilli under a dormant state and extracted mRNA. The intracellular H37Ra in the THP-1 macrophages after treatment with 50 µM $H_2O_2$ was included as a control. First, we monitored the expression levels of *M. tuberculosis* genes such as DosR, *lat*, and *tgs1* which were known to be responsive to mycobacterial dormancy (31, 33, 34). Indeed, their expression was significantly induced in intracellular H37Ra residing in $H_2O_2$-treated RPE cells, which was not clear in $H_2O_2$ treated THP-1 macrophages (Fig. 2B). The qRT-PCR result, growth kinetics, and ROS levels collectively indicated that H37Ra phenotypic dormancy in the $H_2O_2$-treated RPE cells was largely attributed to the greater level of redox stress within RPE cells than that of THP-1 macrophages. As we observed that DosR expression was induced under mycobacterial dormancy within the RPE cell culture system, we monitored 20 selected DosR regulon genes at days 1 and 3 after H37Ra infection (35, 36). As expected, the foregoing DosR regulon gene expression in RPE cells was significantly induced and kinetically matched to that of DosR, *lat*, and *tgs1*, but their responses became clearer at day 3 after H37Ra infection (Fig. 2C; Fig. S2C). Consistent with induced *tgs1* expression, we also observed a greater accumulation of TAG at *M. tuberculosis* cell wall during residing inside the RPE cells than that of THP-1 macrophages (Fig. 2A).

Previous studies showed that host immune cells were immunometabolically blunted to respond to the latent TB infection as *M. tuberculosis* in a dormant state significantly repopulates the glycolipids, especially ligands functioning as a pathogen-associated molecular pattern (37–40). As RPE cells are the main source to secrete inflammatory cytokines in the retina to build up the intraocular immune environment, we monitored

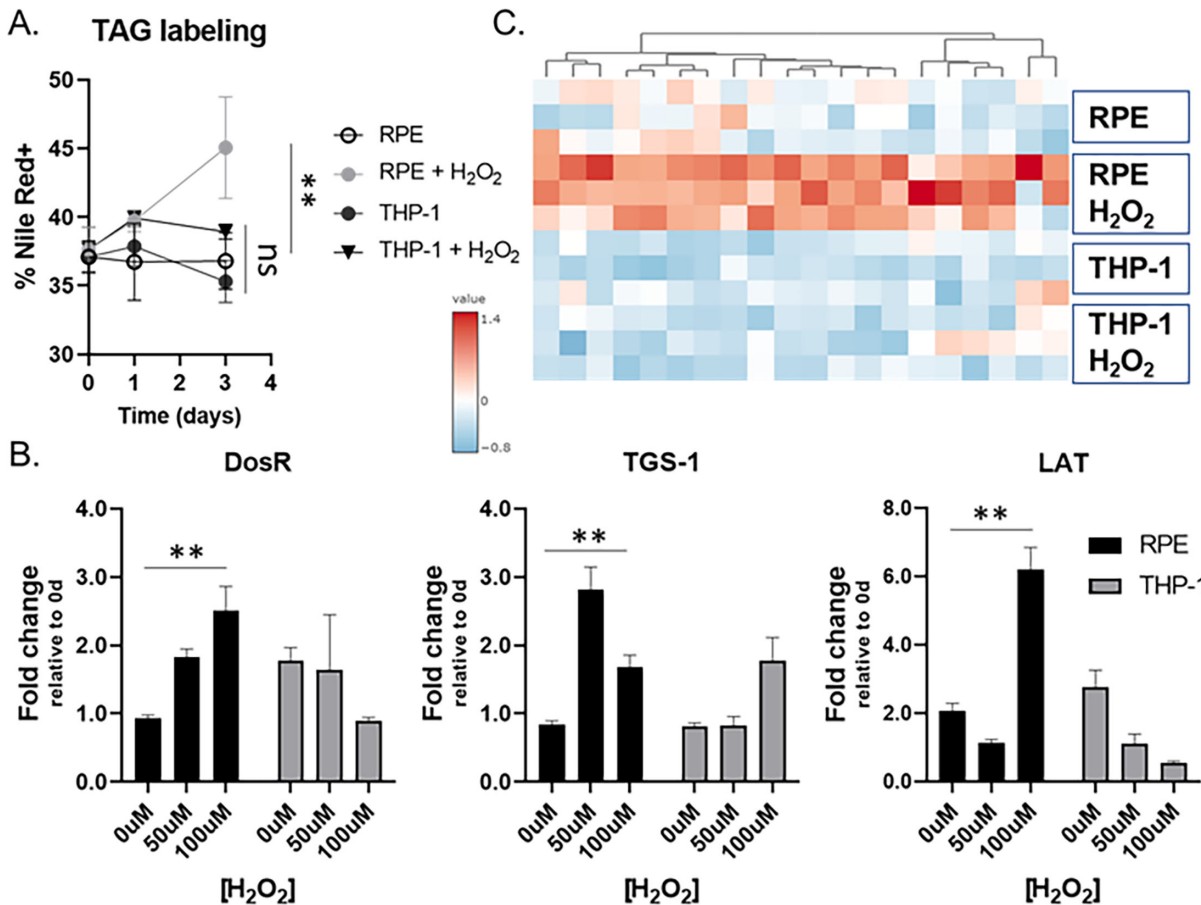

**FIG 2** Metabolic evidence indicating the mycobacterial dormancy in the RPE cells. (A) Accumulation of TAG at *M. tuberculosis* cell wall was monitored after phagocytosed by RPE cells or THP-1 macrophages. (B) Intracellular H37Ra mRNA transcript levels of dormancy and persistence-associated genes following infection of RPE cells (black) or THP-1 macrophages (gray) and treatment with 0, 50, or 100 µM $H_2O_2$ for 1 day. (C) Heatmap depicting the mRNA transcripts of randomly selected DosR regulon genes following infection of RPE cells or THP-1 macrophages with 50 µM $H_2O_2$ treatment for 3 days. **, $P < 0.05$; ns, not significant by Student's *t* test.

the expression levels of proinflammatory cytokine genes such as *il6* and *tnfα* in the $H_2O_2$-treated RPE cells or THP-1 macrophages after infection with H37Ra. We observed greater induction of *il6* and *tnfα* expression in THP-1 macrophages infected with H37Ra than that of RPE cells, supporting our speculation that *M. tuberculosis* bacilli reside in RPE cells prevented from hyperactivation of proinflammatory response of host ocular immune system due to their phenotypic dormancy (Fig. S2A and B). Intriguingly, $H_2O_2$ treatment alone was not sufficient to induce the expression of these cytokines from both cell types, confirming the critical role of mycobacterial dormancy in the interaction between host and pathogen for the *M. tuberculosis* survival benefit during mycobacterial dormancy.

## Metabolomics profile of $H_2O_2$-treated RPE cells for the dormant *M. tuberculosis* infection

To understand the mechanistic bases of $H_2O_2$-treated RPE intracellular milieu behind the mycobacterial dormancy, we set out to identify metabolic states of RPE cells altered by $H_2O_2$ treatment and/or H37Ra infection using LC-MS metabolomics (41, 42). $H_2O_2$-treated THP-1 macrophages with or without H37Ra infection were included as controls. The metabolic changes were determined by comparing the abundance of approximately 430 metabolites of RPE cells and THP-1 macrophages selected from the Metlin database (http://metlin.scripps.edu) with those of conditions with neither $H_2O_2$ treatment nor H37Ra infection (43). Since metabolome extraction was conducted without the step

to separate intracellular H37Ra from the host cells, the detected metabolites could be from the host and/or pathogen. Using Metaboanalyst v5.0, multivariate unbiased clustering analyses identified a subset of metabolites and mapped annotated pathways. The pathway mapping analysis revealed that glutamate metabolism, amino acid metabolism (e.g., alanine, aspartate, ornithine, proline, and aromatic amino acids), urea cycle, and antioxidant metabolism pathways were upregulated when RPE cells were co-stimulated with $H_2O_2$ and H37Ra infection. Reciprocally, β-alanine, arginine, and histidine metabolism pathways were downregulated as compared to those in RPE cells in a resting condition (Fig. S3A through C). The untargeted metabolomics profiles and principal component analysis (PCA) suggested that treatment with 50 μM $H_2O_2$ alone was not sufficient to alter the metabolic networks of RPE cells (Fig. S4A), but additional H37Ra infection profoundly changed the RPE and THP-1 cellular metabolic networks (Fig. 3A and B; Fig.S4B). The targeted lipidomics profiles and PCA, based upon a database containing intermediates in fatty acid metabolism, cholesterol metabolism, and other lipid metabolism pathways, matched the conclusions drawn from the untargeted metabolomics analysis (Fig. S5A and B). Intriguingly, the metabolic networks of THP-1 macrophages between days 1 and 3 after H37Ra infection were not clearly different, indicating that the RPE cells at day 1 and THP-1 metabolic states at both days 1 and 3 were not sufficient to confer the mycobacterial dormancy.

$H_2O_2$-treated RPE metabolic networks were continuously altered as the duration of H37Ra infection time was undergone. However, $H_2O_2$-treated THP-1 metabolic networks were not further altered even after additional stimulation with H37Ra infection (Fig. 3A and B; Fig. S4C). The PCAs indicated that $H_2O_2$-treated RPE metabolic networks at day 3 after H37Ra infection were important metabolic activities required to trigger the formation of a dormant H37Ra phenotypic state. Indeed, the metabolic networks included the enhanced abundance of metabolites involved in the antioxidant metabolism pathways such as glutathione, glutamate metabolism intermediates, glutamyl cysteine, cysteine, glycine, and GSH(Glutathione) disulfide, whose changes were not clear in RPE cells at day 1 after H37Ra infection or THP-1 cells at both days (Fig. 3C and D; Fig. S4D). The induced abundances of antioxidant metabolism pathway intermediates were fully restored by co-treatment with 10 mM thiourea (Fig. S4D). Intracellular ROS level in $H_2O_2$-treated RPE cells was reduced by treatment with 10 mM thiourea, corroborating with the mycobacterial dormancy and growth kinetics (Fig. 1B; Fig. S1B). Collectively, these findings suggested that redox stresses of $H_2O_2$-treated RPE cells were significantly enhanced by *M. tuberculosis* infection at and after day 3, and the intracellular metabolic environment was sufficient to lead *M. tuberculosis* bacilli to enter a dormant state. These findings were also supported by qRT-PCR results of $H_2O_2$-treated RPE cells at day 3 after H37Ra infection and intracellular H37Ra growth kinetics (Fig. 1A, 2B and C).

## Antibiotic tolerance of intracellular *M. tuberculosis* within RPE cells was restored by co-treatment with phosphoenolpyruvate

The mycobacterial dormancy is in most cases accompanied by a high level of antibiotic tolerance against clinically important TB antibiotics such as isoniazid (INH) and rifampicin (RIF) (1, 27, 38, 44). We monitored the antibiotic sensitivity of intracellular H37Ra within $H_2O_2$-treated RPE cells against 5× (minimal inhibitory concentration) MIC-equivalent INH or RIF for a week. Treatment with RIF initially killed the bacilli within RPE cells and THP-1 macrophages by approximately 1.5 $\log_{10}$ CFU during the first 2 days. While the viable number of H37Ra within THP-1 macrophages continued to decrease, the viability of H37Ra within RPE cells was maintained after day 2 post treatment with RIF until day 7 (Fig. 4A, left panel). A greater level of antibiotic tolerance of H37Ra in the RPE cells than that of THP-1 macrophages was also observed after treatment with 5× MIC-equivalent INH (Fig. 4A, right panel). The greater antibiotic tolerance of H37Ra residing in the $H_2O_2$-treated RPE cells seemed to be largely due to its phenotypic dormancy induced by RPE intracellular redox stresses.

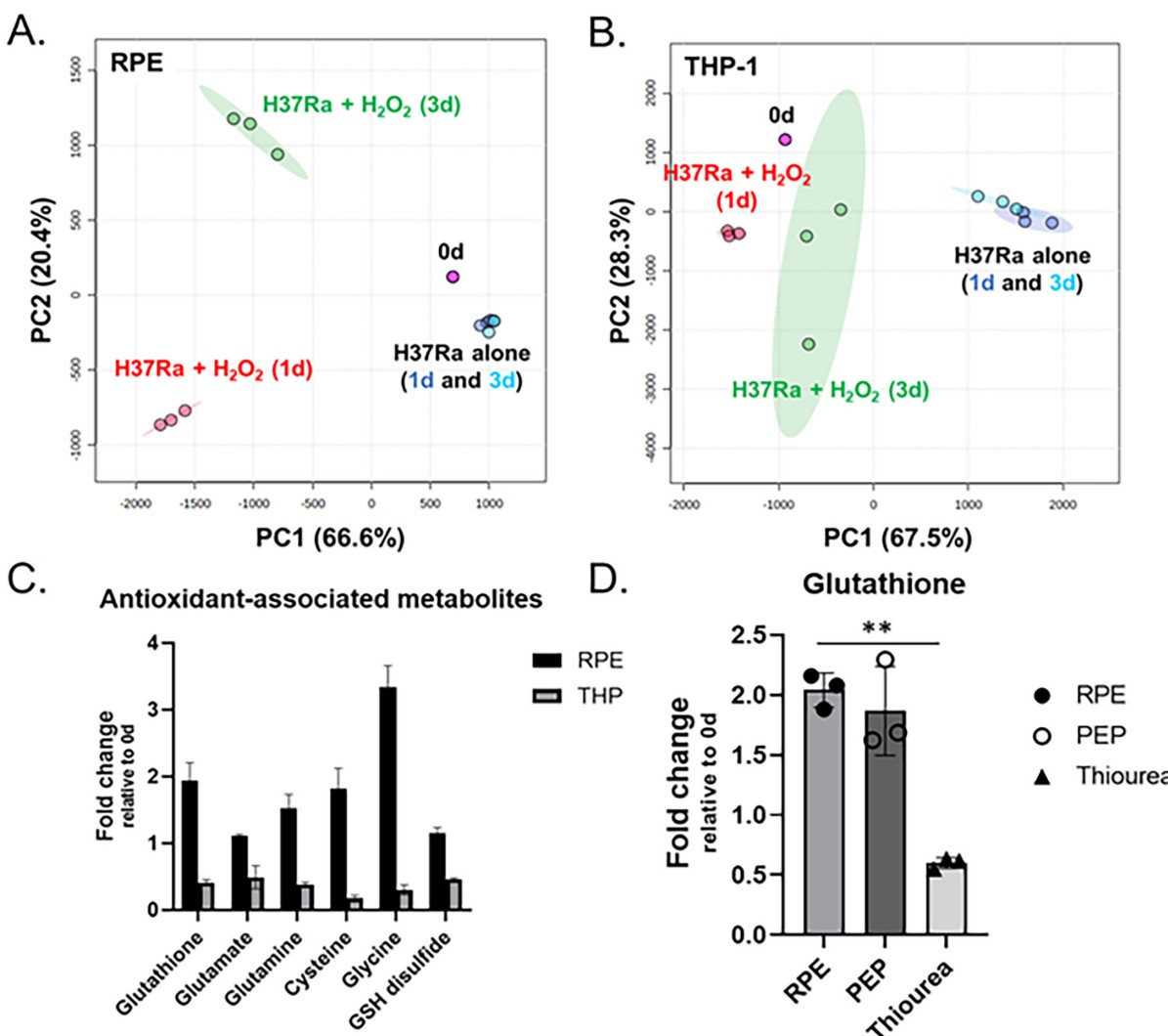

**FIG 3** Intracellular metabolic state of RPE cells after co-treatment with $H_2O_2$ and H37Ra infection. (A and B) PCA of the metabolome profiles of infected RPE cells (A) and THP-1 macrophages (B) with or without treatment with 50 µM $H_2O_2$. Metabolites were collected at days 0, 1, and 3 post-infection. Pink spheres, triplicate samples of cells before infection or treatment with $H_2O_2$. Dark and light blue spheres, infected cells at days 1 and 3 post-infection, respectively. Red and green spheres, infected cells treated with $H_2O_2$ at days 1 and 3 post-infection, respectively. (C) Metabolite abundance of RPE cells and THP-1 macrophages treated with 50 µM $H_2O_2$ for 3 days. Bars are representative of triplicate samples and show average fold change relative to those at day 0. (D) Glutathione abundance in infected RPE cells following treatment with 50 µM $H_2O_2$ and supplementation of 10 mM PEP or thiourea. Data points show the average of experimental triplicates ± SEM. **, $P < 0.05$ by Student's $t$ test.

We recently studied the semi-untargeted metabolomics profile of H37Rv phenotypic dormancy during adaptation to hypoxia and showed that central carbon metabolism pathways including glycolysis, TCA (tricarboxylic acid) cycle, and pyruvate metabolism pathways were highly altered in dormant H37Rv bacilli (37, 45, 46). Targeted metabolomics analysis showed a nearly complete depletion of PEP, the most downstream metabolite in the *M. tuberculosis* glycolysis. Exogenous supplementation with PEP significantly recovered the hypoxic *M. tuberculosis* growth compared to that without PEP, thereby improving antibiotic susceptibility to first-line TB antibiotics, INH and RIF. Thus, we first tested if treatment with PEP to $H_2O_2$-treated RPE cells prevented intracellular H37Ra within the cells from entering to a dormant state. Consistent with previous reports, supplementation with PEP significantly improved the growth rates of intracellular H37Ra within $H_2O_2$-treated RPE cells (Fig. S6A), indicating that externally supplemented PEP accessed the intracellular H37Ra and modulated its growth kinetics and

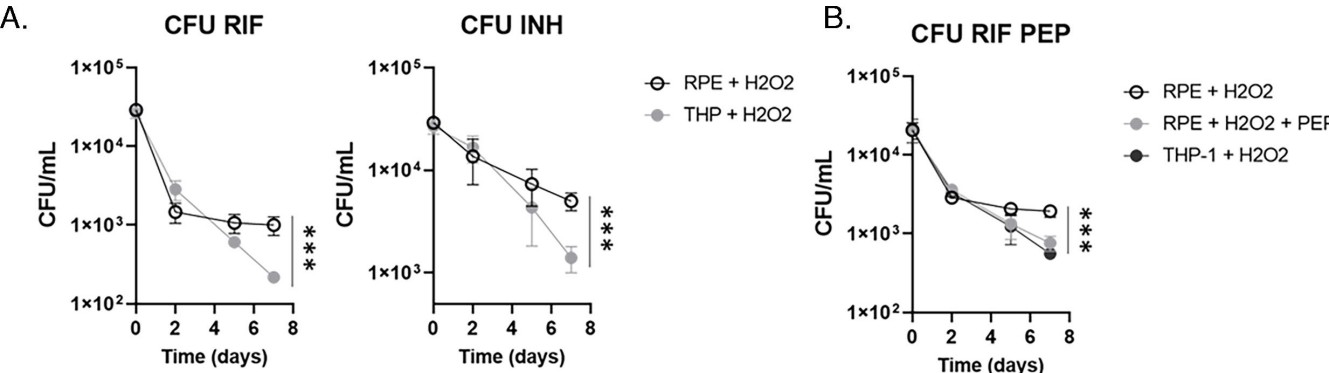

**FIG 4** Antibiotic tolerance of intracellular *M. tuberculosis* and its synthetic lethality. (A) CFU viability of intracellular H37Ra within RPE cells or THP-1 macrophages following co-treatment with 50 µM $H_2O_2$ and 5× MIC-equivalent RIF (left panel) or INH (right panel). (B) The effect of supplementing with 10 mM PEP on the CFU viability of intracellular H37Ra following exposure to 5× MIC-equivalent RIF or INH. ***, $P < 0.001$ by ANOVA.

metabolic activities. The growth restored by PEP supplementation was not by altering RPE metabolic networks but by directly modulating H37Ra bacterial metabolic networks (Fig. S6B). We next observed that intracellular H37Ra within $H_2O_2$-treated RPE cells was synthetically lethal to 5× MIC-equivalent RIF by co-treatment with 10 mM PEP, resulting in restoring the RIF-mediated bactericidal effect to a level similar to that of intracellular H37Ra within $H_2O_2$-treated THP-1 macrophages or RPE cells in a resting condition (Fig. 4A and B). The PEP-mediated improved antibiotic susceptibility of H37Ra residing in $H_2O_2$-treated RPE cells indicated that mycobacterial dormancy of *M. tuberculosis* within ocular tissues was associated with its metabolic adaptation and can be restored by reverse metabolic modulation by co-treatment with PEP. The metabolic modulation strategy has been intensively studied (45, 47–50) and thus considered a potential method in designing novel TB antibiotic regimen to overcome mycobacterial dormancy-mediated antibiotic tolerance often observed in OTB cases. We also tested the foregoing PEP-mediated synthetic antibacterial effects using Linage 2 W-Beijing HN878, a clinical strain known to harbor the highest risk of developing MDR (multidrug resistance) cases worldwide (51, 52). We observed that intracellular HN878 within $H_2O_2$-treated RPE cells was tolerant to RIF treatment, but the RIF sensitivity was significantly improved by co-treatment with 10 mM PEP (Fig. S7). This therapeutic strategy should be efficient to treat OTB patients as the drug delivery toward the site of infection may be much convenient compared to that within the pulmonary organs. Recent studies identified that RPE cells serve as a major niche that phagocytoses invading Mtb bacilli in OTB cases and the Mtb bacilli reside in a dormant and drug-tolerant state (22). Our result collectively provides the metabolic and biochemical bases why intracellular Mtb bacilli within RPE cells maintain the phenotypic dormancy and high levels of antibiotic tolerance. This study will be a great opportunity for mass eradication of *M. tuberculosis* infection in the ocular tissues by synthetically waking up the dormant *M. tuberculosis* bacilli.

## MATERIALS AND METHODS

### Bacterial strains, culture conditions, and chemicals

*M. tuberculosis* lineage 2 W-Beijing HN878 was cultured in a biosafety level 3 facility, and H37Ra was cultured in a biosafety level 2+ facility at 37°C in Middlebrook 7H9 broth (m7H9) or on Middlebrook 7H10 agar (m7H10; Difco) supplemented with 0.2% glucose, 0.04% tyloxapol (broth only), 0.5 g/L BSA (Bovine Serum Albumin), and 0.085% NaCl. We have complied with all relevant ethical regulation for work with *M. tuberculosis* clinical strains such as HN878.

## PPE and THP-1 cell cultures

Human fetal RPE cells were purchased from ATCC. Passage 2 cells were transferred to 75 mL flasks and grown in confluence in Dulbecco Modified Eagle Medium (DMEM; Lonza BioWhittaker) containing 10% heat-inactivated fetal bovine serum (Gibco, Life Technologies), l-glutamine (1 mM), hydroxyethyl piperazineethanesulfonic acid buffer (10 mM), and penicillin G sodium and streptomycin sulfate (50 U/mL). For both RPE and THP-1 cells, the medium was changed twice each week. During the medium change, the cell cultures were evaluated by phase-contrast microscopy using trypan blue, and the number of nonviable cells was less than 5% in all experiments before *M. tuberculosis* was infected. The cells were transferred into 96-, 12-, or 6-well plates for intracellular *M. tuberculosis* load analysis.

Acute monocytic leukemia cell line (THP-1) was cultured in Roswell Park Memorial Institute (RPMI) 1640 medium containing 10% heat-inactivated fetal bovine serum and penicillin G sodium and streptomycin sulfate (50 U/mL). Adhesion of THP-1 to culture plates was attained by adding 20 ng/mL of phorbol 12-myristate 13 acetate (PMA) to each well. The slide or the plates were washed with PBS (Phosphate Buffered Saline) (37°C) twice before seeding. The THP-1 cells were plated in 96-, 12-, or 6-well plates; they were left still for 12 hours overnight with PMA-containing medium and then washed with 37°C PBS to remove nonadherent cells. The THP-1 cells were evaluated daily for morphologic changes, including cytoplasmic projections.

## Mtb H37Ra or HN878 infection

The H37Ra or HN878 was cultured in Middlebrook 7H9 broth (BD Biosciences) to mid-logarithmic phase (optical density approximately 0.5). The culture was then pelleted, resuspended in an equal volume of PBS, and centrifuged at 800 RPM for 12 minutes to generate a single-cell suspension. One milliliter of aliquots containing $5 \times 10^7$ cells/mL was stored at –80°C until the morning of infection. RPE and THP-1 cell cultures were seeded at a density of $1 \times 10^5$ cells/well in a 96-well plate and were infected with an MOI of 10:1. After a 4-hour exposure of cells to *M. tuberculosis*, the attached cells in the culture wells were washed with culture medium to remove nonadherent Mtb. A complete medium (DMEM for RPE and RPMI1640 for THP-1) containing gentamycin (25 µg/mL) was then added to the culture wells for 24 hours to kill the remaining extracellular Mtb attached to the cell surface. At this time and beyond, the medium was replaced with an antibiotic-free medium.

## Cell viability assessment

Cells were seeded at a density of $5 \times 10^5$ cells/well in a 24-well plate and allowed to adhere for 24 hours. They were then washed twice with 37°C PBS to remove non-adherent cells, and a complete medium containing $H_2O_2$ (0–150 µM) was added to the culture wells for 24 hours. Cell supernatants were collected, and the number of viable cells was counted by phase-contrast microscopy using trypan blue exclusion.

## *Ex vivo* CFU enumeration assay

Enumeration of viable intracellular Mtb was accomplished by lysing infected cells with 100 µL 0.5% Triton X-100. After incubation at 37°C for 5 minutes, cell lysates were harvested, and released bacilli were enumerated by plating serial dilutions of the lysate on m7H10 agar. All cultures were performed in triplicate. Agar plates were cultivated in a 37°C humidified incubator with 5% $CO_2$ for 3–4 weeks before determining the CFU.

## *In vitro* CFU enumeration assay

Mtb viability was determined using liquid cultures manipulated under experimentally identical conditions for metabolomics and qRT-PCR profiling, which we previously demonstrated to be microbiologically similar. CFUs were determined by plating serial

dilutions on m7H10 after incubating for at least 3–4 weeks at 37°C. CFU enumeration assays were conducted in two independent triplicates.

## Metabolite extraction for LC-MS analysis

Cells were seeded at a density of $5 \times 10^5$ cells in a 6-well plate and allowed to adhere for 24 hours. Cells were then infected as described previously, and culture media was replaced with complete media containing $H_2O_2$ (50 µM), PEP (10 mM), or thiourea (10 mM) as needed. Metabolites were extracted at 1 or 3 days post-infection (43, 53). Cells were washed three times with PBS and metabolically quenched with 500 µL ice-cold 40:40:20 (acetonitrile:methanol:water). Adherent cells were mechanically detached, and metabolites were extracted by lysis with 0.1 mm zirconia beads in a Precellys tissue homogenizer for 1 minute under continuous cooling at or below 2°C. Lysates were clarified by centrifugation, and the residual protein content of metabolite extracts was determined (BCA protein assay kit, Thermo Scientific) to normalize the samples to cell biomass.

## LC-MS for metabolomics profiling

LC-MS differentiation and detection of each metabolite were performed with an Agilent Accurate Mass 6230 TOF coupled with an Agilent 1290 Liquid Chromatography system using a Cogent Diamond Hydride Type C column (Microsolv Technologies, Long Branch, NJ, USA) with solvents and configuration as previously described (38, 46). An isocratic pump was used for continuous infusion of a reference mass solution to allow mass axis calibration. Detected ions were classified as metabolites based on unique accurate mass-retention time identifiers for masses showing the expected distribution of accompanying isotopologues. Metabolites were analyzed using Agilent Qualitative Analysis B.07.00 and Profinder B.07.00 software (Agilent Technologies, Santa Clara, CA, USA) with a mass tolerance of <0.005 Da. Standards of authentic chemicals of known amounts were mixed with bacterial lysates and analyzed to generate the standard curves used to quantify metabolite levels. All data obtained by metabolomics profiling were the average of at least two independent triplicates. Bioinformatics analysis was carried out using MetaboAnalyst v5.0 (www.metaboanalyst.ca), which is a web-based available software for processing metabolomics data, and pathway mapping was performed on the basis of annotated human metabolic pathways available in the Kyoto Encyclopedia of Genes and Genomes pathway database. Metabolomics data were analyzed by statistical analysis. The clustered heatmap and hierarchical clustering trees were generated using Cluster 3.0 and Java Tree View 1.0. A univariate statistical analysis involving an unpaired *t* test was used to identify significant differences in the abundance of metabolites between each group.

## RNA extraction and qRT-PCR

Cells were seeded at a density of $1 \times 10^5$ cells/well in a 12-well plate and infected as previously described (54). Total RNA from Mtb-infected RPE or THP-1 cells was extracted using TRIzol Reagent (Sigma-Aldrich) and mechanical lysing with 0.1 mm zirconia beads in a Percellys tissue homogenizer. Lysates were clarified by centrifugation, and TRIzol supernatant was removed and used for RNA extraction. RNA was isolated using a Qiagen RNA extraction kit. RNA concentrations were determined using a Nanodrop, and qRT-PCR was performed using an iQ SYBR-Green Supermix (Bio-Rad) and Mastercycler ep Realplex 2 (Eppendorf). Data were normalized by GAPDH (Glyceraldehyde 3-phosphate dehydrogenase) or *sigA* expression level, and all primers were designed using GenScript primer design software. All primers and sequences are available in Table 1.

## ROS measurement

Intracellular ROS of RPE and THP-1 cells was determined by previously established protocols (45, 55). The ROS was monitored by measuring the changes in the florescence

**TABLE 1** qRT-PCR primers and sequences

|  | Forward | Reverse |
|---|---|---|
| DosR | CCGAATGTTCCTAGCCGAAA | TTCAACTCCGTCGCGAATAC |
| lat | CTGGACATAGTGCTCGATCTG | AGGAGGCAACGAATGTGAA |
| tgs1 | GGGTTTCTCAAGGCAGAAGA | GTTGAGCGAGCGACGATAA |
| Rv0079 | AAACCGGTCGTGCTAAGG | GAACAAATGCACGTCGTAGTC |
| Rv0080 | GAAGCCGACGACCTTGAT | GGTGTCCATCGCCATGTT |
| Rv0081 | CGTTCGGTCGGTGAGTTG | GGTGCGGCAATCGAATAGA |
| Rv0082 | GGGTGTGAGGTGGAGATTT | GTCACCAACAACGCATCG |
| Rv0083 | CACGTCCGCGCTGTATG | CATGTTCTCGGTCGTCGAATAG |
| Rv0569 | GACCACCGAGGGTTGATTATT | GAATCACCGTCGCCACAT |
| Rv0570 | TGTTTCTCGACACGATCAATAGG | GCCGAGATTACATGACTCGTAAG |
| Rv0571 | GGTTTGCGACGCTGTTATTC | AAGCGAGCAGCTCAATGT |
| Rv0572 | CGCGGATTCTGGTCTTCAC | TTGCCGTCCTGGAAGTAGTA |
| Rv0574 | CGGATGCACCCGGATAAC | TGGTAGCCGAAATCGAGAATG |
| Rv1733 | GTGATCGACAGCAACACGA | TCACCGCTGCGTTCTATTC |
| Rv1734 | GTCATGACGACCGTGCT | CGCGTCAGCTCGTTGAT |
| Rv1736 | TGCACACCATCTCCACATAC | CCGATTAGCTCCACGAACC |
| Rv1737 | CTTCGTGATGCACCCTACTT | CGCGTACAGAAACGACATCT |
| Rv1738 | AAGGAATTGGTGGGTGTTGG | ACCTTCAACATTCGCTTCCC |
| Rv1812 | CTTCGGATCCCTTGGTATGTC | GCCCTGCGATACCACTTT |
| Rv1813 | CAGAGCAAGTCGCACTAGAAA | TGGTATTTCGAGCCGTTGTAG |
| Rv1996 | CGAATGGAGAAACCTCGAAGA | ATCGCACACCACGACTTT |
| Rv2623 | CCACGGTCCACAGTGAAAT | GACAACCCACGACCATCAG |
| SigA | ACGAAGACCACGAAGACCTCGAA | GTAGGCGCGAACCGAGTCGGCGG |
| il-6 | AAGCCAGAGCTGTGCAGATGAGTA | TGTCCTGCAGCCACTGGTTC |
| tnfα | CCCAGGGACCTCTCTCTAATC | ATGGGCTACAGGCTTGTCACT |
| gapdh | GTGGTCTCCTCTGACTTCAACA | CTCTTCCTCTTGTGCTCTTGCT |
| DosR | CCGAATGTTCCTAGCCGAAA | TTCAACTCCGTCGCGAATAC |

of the ROS-sensitive fluorophore $H_2DCF$-DA. Cells were initially seeded at a density of $1 \times 10^6$ cells/mL in a 12-well plate and then treated with various concentrations of $H_2O_2$. After incubating at 37°C for 0, 1, and 3 days, 10 µM of $H_2DCF$-DA dye was added and incubated at 37°C for an additional 30 min. The fluorescence was measured by a microplate reader at 485/535 nm. After measuring the ROS values, the amount was normalized to the total number of cells and expressed relative to those at day 0.

## TAG staining and flow cytometry assay

*M. tuberculosis* cells, recovered from RPE cells or THP-1 macrophages following treatment with $H_2O_2$ for 1 or 3 days, were concentrated by centrifugation and stained with Auramine-O and Nile Red at 1:100 dilution from a stock of 100 µg/mL in methanol. Cells were mixed and incubated for 15 min in the dark being inverted every 3 min for mixing. Cells were washed with PBS twice, and labeling intensity was monitored by Invitrogen Attune Flow Cytometer (31).

## Statistical analysis

All experiments including metabolomics and qRT-PCR were conducted in triplicate and repeated at least two times unless otherwise described. The results shown represented mean ± SEM. Statistical analyses were conducted Prism software (GraphPad 9). To determine statistical significance, we employed two-tailed unpaired Student's *t* tests for comparing two groups and ANOVA tests for comparing more than two groups. We considered $P < 0.05$ statistically significant.

## ACKNOWLEDGMENTS

We thank all members of Hyungjin Eoh lab members for their help throughout the work described in this report. This work was supported by NIH grant R01 AI168088.

R.L., L.R., and E.H. designed research studies. R.L., D.J.N., L.R., and L.J.J. conducted experiments. R.L., D.J.N., L.R., and E.H. analyzed data. K.N. and A.H. provided materials. R.L., D.J.N., R.N., and E.H. wrote the manuscript.

## AUTHOR AFFILIATIONS

[1]Molecular Microbiology and Immunology, Keck School of Medicine, University of Southern California, Los Angeles, California, USA
[2]Roski Eye Institute, Keck School of Medicine, University of Southern California, Los Angeles, California, USA
[3]School of Pharmacy, Sungkyunkwan University, Suwon, South Korea

## AUTHOR ORCIDs

Hyungjin Eoh  http://orcid.org/0000-0001-8774-6400

## FUNDING

| Funder | Grant(s) | Author(s) |
| --- | --- | --- |
| HHS | NIH | National Institute of Allergy and Infectious Diseases (NIAID) | R01 AI168088 | Hyungjin Eoh |

## DATA AVAILABILITY

All data generated or analyzed during this study are included in this published article and its supplemented files. The metabolomics raw datasets were deposited in Metabo-Lights. The accession number is MTBLS9841.

## ADDITIONAL FILES

The following material is available online.

### Supplemental Material

**Supplemental material (Spectrum00788-24-S0001.pdf).** Fig. S1 to S7.

### Open Peer Review

**PEER REVIEW HISTORY (review-history.pdf).** An accounting of the reviewer comments and feedback.

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
