## [Reviewer comments · Microbiology Spectrum]

Microbiology Spectrum

***Mycobacterium* dormancy and antibiotic tolerance within the retinal pigment epithelium of ocular tuberculosis**

Rachel Liu, Joshua Dang, Rho Eun Lee, Jae-Jin Lee, Niranjana Kesavamoorthy, Hossein Ameri, Narsing Rao, and Hyungjin Eoh

Corresponding Author(s): Hyungjin Eoh, University of Southern California Keck School of Medicine

Review Timeline:

Submission Date:	March 28, 2024
Editorial Decision:	April 21, 2024
Revision Received:	May 3, 2024
Accepted:	May 23, 2024

Editor: Prabakaran Narayanasamy

Reviewer(s): The reviewers have opted to remain anonymous.

Transaction Report:

DOI: <https://doi.org/10.1128/spectrum.00788-24>

Re: Spectrum00788-24 (Mycobacterium dormancy and antibiotic tolerance within the retinal pigment epithelium of ocular tuberculosis)

Dear Dr. Hyungjin Eoh:

Thank you for the privilege of reviewing your work. Below you will find my comments, instructions from the Spectrum editorial office, and the reviewer comments.

Nice work and we recommend to do the required modifications suggested by our Reviewers with point-by-point response..

Revision Guidelines

Sincerely,
Prabakaran Narayanasamy
Editor
Microbiology Spectrum

Reviewer #1 (Comments for the Author):

The manuscript, Mycobacterium dormancy and antibiotic tolerance within the retinal pigment epithelium of ocular tuberculosis by Dr. Hyungjin Eoh, describes important metabolic changes in the retinal pigment epithelium (RPE) cell that serve as a reservoir of dormant Mtb. The authors provide evidence of these changes through biochemical and phenotypic analysis data. I have several comments:

1. The authors show in Figure 1A that Mtb in RPE without H₂O₂ do not enter dormancy, but instead exhibits vegetative growth. The authors should provide a more detailed explanation or discussion of how the H₂O experiment is relevant to clinical settings, particularly in Mtb-infected hosts without external H₂O supplementation, where Mtb in the RPE of the infected host would likely be in an actively growing state.
2. In Figure 3, the authors conducted metabolic profiling on Mtb-infected RPE but did not explain how they excluded metabolites from the intracellular bacilli (Mtb). The authors should provide an explanation or discussion of this issue, as the total metabolites from Mtb-infected RPE may include contributions from both organisms.
3. In Figure 2C, I strongly recommend the authors add labels of genes at each box, and include p-values for the selected genes, as the fold changes of those genes do not seem to be significant.
4. Lastly, the first paragraph of the Results and Discussion section (page 6) seems redundant. I recommend combining it with the Introduction.

Reviewer #2 (Comments for the Author):

In their manuscript, Liu et al. present a study elucidating the metabolic characteristics of retinal pigment epithelium (RPE) cells interacting with *Mycobacterium tuberculosis* (Mtb), juxtaposed with THP-1 cells. Despite the intriguing results showcased by the authors, this review raises several pertinent concerns regarding the rationale and interpretation of the study.

Major concerns:

1. The utilization of Mtb strain H37Ra warrants scrutiny due to significant variances in virulence, dormancy, and pathogenesis between virulent and avirulent strains. While the inclusion of Mtb strain HN878 in select experiments is noted, the primary conclusions stem from Mtb H37Ra. Therefore, a comprehensive discussion on the disparities between virulent and avirulent Mtb strains is imperative.
2. The focus on polar metabolites in the study contrasts with the predominantly lipid-associated nature of Mtb dormancy. Hence, an examination of lipid accumulation and alterations is warranted to augment the investigation.

Minor concerns:

1. The title's broad scope necessitates the inclusion of representative findings to enhance specificity.
2. In the abstract, a shift towards highlighting major findings over extensive background elucidation is recommended for conciseness.
3. Exploration into evidence supporting the reactivation of ocular TB from dormant Mtb bacilli would enrich the discussion.
4. The absence of detectable ROS levels in THP-1 cells post-H₂O₂ treatment, as observed in Supplementary Figure 1, warrants clarification.
5. Discrepancies in Mtb colony-forming units (CFUs) between Figure 1 and Supplementary Figure 1A merit elucidation, particularly concerning implications for Mtb dormancy.
6. Comparative discussion of this study's findings with existing literature investigating the link between RPE cells and Mtb dormancy would provide valuable context.
7. Given the study's utilization of a small sample size, ensuring appropriate statistical methodologies such as the Mann-Whitney test versus Student t-test is paramount to bolster the robustness of the findings.

In response to all reviewers, we provide a considerable amount of new experimental data that bolster the importance of RPE cellular intracellular environments on mycobacterial dormancy and antibiotic tolerance frequently seen in OTB patients. In response to reviewers below, we have referred to the new figure numbers of the revised manuscript.

Editor:

Nice work and we recommend to do the required modifications suggested by our Reviewers with point-by-point response.

Thank you for the comment.

Reviewer #1 :

1. The authors show in Figure 1A that Mtb in RPE without H₂O₂ do not enter dormancy, but instead exhibits vegetative growth. The authors should provide a more detailed explanation or discussion of how the H₂O₂ experiment is relevant to clinical settings, particularly in Mtb-infected hosts without external H₂O₂ supplementation, where Mtb in the RPE of the infected host would likely be in an actively growing state.

Introduction section includes new sentence and reference:

“The vasculature at the choroid system carries high oxygen tension passing through the RPE layer towards the retina. As RPE cells harbor a large number of mitochondria, the mitochondrial complexes generate great levels of oxidative stresses by producing superoxide, H₂O₂, and hydroxyl radical. The intrinsic environment within RPE cells thus is high level of oxidative stresses and RPE cells have an important role in meeting the metabolic demand.” (page 5).

Result section also includes new sentence:

“The growth kinetics revealed that the mycobacterial dormancy state in the RPE cells needs the environmental redox stresses” (page 6).

2. In Figure 3, the authors conducted metabolic profiling on Mtb-infected RPE but did not explain how they excluded metabolites from the intracellular bacilli (Mtb). The authors should provide an explanation or discussion of this issue, as the total metabolites from Mtb-infected RPE may include contributions from both organisms.

Thank you for the comment. We employed this method as previously conducted elsewhere (Chandra et al. 2022, PMID: 35104812) and included the following sentence: “Since metabolome extraction was conducted without the step to separate intracellular H37Ra from the host cells, the detected metabolites could be from host and/or pathogen.” (page 8)

3. In Figure 2C, I strongly recommend the authors add labels of genes at each box, and include p-values for the selected genes, as the fold changes of those genes do not seem to be significant.

Done. New Supplement Fig. 2C included labels of genes and p-values.

4. Lastly, the first paragraph of the Results and Discussion section (page 6) seems redundant. I recommend combining it with the Introduction.

Thank you for the comment. Done.

Reviewer #2 :

Major concerns:

1. The utilization of Mtb strain H37Ra warrants scrutiny due to significant variances in virulence, dormancy, and pathogenesis between virulent and avirulent strains. While the inclusion of Mtb strain HN878 in select experiments is noted, the primary conclusions stem from Mtb H37Ra. Therefore, a comprehensive discussion on the disparities between virulent and avirulent Mtb strains is imperative.

Thank you for the comment.

H37Ra is a laboratory-conditioned, avirulent mycobacterial strain that has been used as a surrogate for pathogenic *M. tuberculosis* property in many studies. This strain is a good surrogate to test for interacting with host metabolism. Also, our previous study (Nazari et al. 2014, PMID: 24723139) used H37Ra to elucidate the bacterial phenotype and mycobacterial dormancy within the RPE cells. We understand the reviewer's concerns, but the focus of this study resides in elucidating the intracellular metabolic environment within RPE cells that is associated with mycobacterial dormancy and antibiotic tolerance as compared to the macrophages. To overcome the limitation arising from the strain that we used, we also conducted experiments including ex vivo infection, mycobacterial dormancy, and antibiotic synergy using HN878 pathogenic clinical strain and validated that our findings using H37Ra were well applicable to the pathogenic mycobacterial bacilli within RPR cells.

2. The focus on polar metabolites in the study contrasts with the predominantly lipid-associated nature of Mtb dormancy. Hence, an examination of lipid accumulation and alterations is warranted to augment the investigation.

Thank you for the comment.

We conducted the targeted lipidomics analysis using database containing fatty acid metabolism, cholesterol metabolism, and many other lipid metabolisms (Fig. S5). The PCA analysis supported the conclusion (Fig. 3A, B, Fig. S4A-C) drawn from our metabolomics analysis results. New result was included in text (page 8).

Minor concerns:

1. The title's broad scope necessitates the inclusion of representative findings to enhance specificity

Thank you for the comment.

Microbiology, metabolomics, and antibiotic susceptibility tests in this study collectively indicate that mycobacterial bacilli in the RPE cells reside in a dormant state with accompanied antibiotic tolerance. Thus, we believe that current title well fits to describe the findings of this manuscript.

2. In the abstract, a shift towards highlighting major findings over extensive background elucidation is recommended for conciseness.

Thank you for the comment. Done.

3. Exploration into evidence supporting the reactivation of ocular TB from dormant Mtb bacilli would enrich the discussion.

Thank you for the comment. We described that extrapulmonary TB including OTB became pathogenic by the opportunistic reactivation; “Conventional TB chemotherapy struggles to eradicate the bacilli effectively, and thus surviving bacilli can regrow when the antibiotic effects wane or when resistant mutants emerge.” (page 4).

We also included new sentence with references as follows: “RPE cells are responsible for the immune privilege of ocular tissues, making them a primary niche where invading pathogens reside. Indeed, previous studies investigating the cases of OTB uveitis have indicated that *M. tuberculosis* mostly localizes within the RPE cells followed by the pathogenic reactivation often when host’s immune system is compromised.” (page 5)

4. The absence of detectable ROS levels in THP-1 cells post-H₂O₂ treatment, as observed in Supplementary Figure 1, warrants clarification.

Using a pilot test, we identified an optimal condition where RPE oxidative stress was significantly greater than that of THP-1 macrophages.

5. Discrepancies in Mtb colony-forming units (CFUs) between Figure 1 and Supplementary Figure 1A merit elucidation, particularly concerning implications for Mtb dormancy.

As described in page 6, Fig. S1A CFU data (left panel) showed the Mtb growth kinetics in an *in vitro* culture condition not in an intracellular environment.

6. Comparative discussion of this study’s findings with existing literature investigating the link between RPE cells and Mtb dormancy would provide valuable context.

We included the new sentence in page 10 by describing the strengths and the weaknesses of prior researches.

“Recent studies identified that RPE cells serve as a major niche that phagocytoses invading Mtb bacilli in OTB cases and the Mtb bacilli reside in a dormant and drug tolerant state. Our result collectively provides the metabolic and biochemical bases why intracellular Mtb bacilli within RPE cells maintain the phenotypic dormancy and high levels of antibiotic tolerance. This study will be a great opportunity for mass eradication of *M. tuberculosis* infection in the ocular tissues by synthetically waking up the dormant *M. tuberculosis* bacilli.”

7. Given the study’s utilization of a small sample size, ensuring appropriate statistical methodologies such as the Mann-Whitney test versus Student t-test is paramount to bolster the robustness of the findings.

We included at least three biological and two technical replicates. Using the replicate data, we used Student t-test and ANOVA statistical methods depending upon the type of experiments.

Re: Spectrum00788-24R1 (Mycobacterium dormancy and antibiotic tolerance within the retinal pigment epithelium of ocular tuberculosis)

Dear Dr. Hyungjin Eoh:

Your manuscript has been accepted, and I am forwarding it to the ASM production staff for publication. Your paper will first be checked to make sure all elements meet the technical requirements. ASM staff will contact you if anything needs to be revised before copyediting and production can begin. Otherwise, you will be notified when your proofs are ready to be viewed.

Sincerely,
Prabakaran Narayanasamy
Editor
Microbiology Spectrum

Reviewer #1 (Comments for the Author):

All concerns that I raised in the previous version of the manuscript have been addressed successfully.

Reviewer #3 (Comments for the Author):

None